# Non-Targeted Metabolomics Reveals the Effects of Different Rolling Methods on Black Tea Quality

**DOI:** 10.3390/foods13020325

**Published:** 2024-01-19

**Authors:** Shuya Yang, Sujan Pathak, Haiyan Tang, De Zhang, Yuqiong Chen, Bernard Ntezimana, Dejiang Ni, Zhi Yu

**Affiliations:** 1National Key Laboratory for Germplasm Innovation & Utilization of Horticultural Crops, College of Horticulture & Forestry Sciences, Huazhong Agricultural University, Wuhan 430070, China; yangshuya777@163.com (S.Y.); sujanp057@gmail.com (S.P.); zdybfq@163.com (D.Z.); chenyq@mail.hzau.edu.cn (Y.C.); bernardntzmn@gmail.com (B.N.); nidj@mail.hzau.edu.cn (D.N.); 2School of Horticulture and Landscape Architecture, Hubei Vocational College of Bio-Technology, Wuhan 430070, China

**Keywords:** black tea, rolling time, aroma, taste, non-targeted metabolomics, sensory evaluation

## Abstract

A non-targeted metabolomics approach and sensory evaluation, coupled with multivariate statistical analysis, systematically uncover the impact of the rolling time on the quality parameters of black tea. GC-MS analysis reveals that a moderate extension of rolling time favorably contributes to the accumulation of characteristic aroma components in black tea. The volatile components reach their highest concentration in black tea samples processed during an 80-min rolling period. UHPLC-Q-TOF/MS analysis demonstrates a substantial decrease in the contents of catechins and flavonoids with an increase in rolling time. Simultaneously, the production of theaflavins, coupled with the degradation of green bitterness volatiles (GBVs), significantly contributes to the formation of endogenous aroma components in black tea. These findings underscore the close relationship between rolling time control and black tea quality, emphasizing that a moderate extension of the rolling time fosters the development of improved black tea flavor quality. The comprehensive quality evaluation indicates that the optimal duration is 80 min. However, the initial 0 to 20 min of rolling is a crucial phase for the genesis and transformation of black tea quality. This study offers valuable insights into the influence of rolling time on black tea quality, potentially enhancing future studies of rolling technology. It provides theoretical guidelines for optimizing the processing of Gongfu black tea.

## 1. Introduction

Tea, an invigorating and revitalizing beverage, is crafted through an infusion of processed and dried leaves from the *Camellia sinensis* L. (O) Kuntze plant [1]. It is the favorite drink for approximately three-quarters of the global population [2]. Based on variations in manufacturing processes and quality attributes, tea is categorically divided into six types: green tea, white tea, oolong tea, yellow tea, dark tea, and black tea. Notably, black tea is the most widely produced and consumed type worldwide [3]. Unlike green tea (non-fermented) and oolong tea (semi-fermented), black tea is a fully fermented tea variety that has a dark brown color and a sweet aroma. The processing of black tea mainly involves withering, rolling, fermentation, and drying steps [4]. Of these steps, the rolling process is particularly important. Rolling is considered a pre-fermentation step, as it is the first step that facilitates the oxidation process in tea leaves [5]. The external mechanical force used during rolling breaks down the leaf cells, allowing the polyphenols to come into contact with enzymes such as polyphenol oxidase enzymes (PPO) and peroxidase (POD) in the cytoplasm. These enzymes produce pigment compounds such as theaflavins and theobromine. Additionally, the cytosol from the damaged leaf cells overflows and attaches to the surface of the tea leaves. When dried, this component gives the tea its dark color and enhances its flavor [5].

Furthermore, rolling is a key step in the process of converting tea leaves into strip form to produce black tea. These strips can be obtained either manually by hand or through mechanized methods [6]. On one hand, the mechanical force applied during rolling helps to break down the cells of tea leaves, facilitating the desired rate of formation. Additionally, the tea juice that overflows during rolling adheres to the leaf surface, and the starch and pectin substances present in the juice contribute to enhancing the color and oiliness of the dried tea. As a result, the tea appears moist even after drying [7]. Therefore, the control of the degree of rolling becomes a particularly important consideration in the process of black tea rolling. Despite being one of the most important steps in black tea processing, researches and studies focusing on this particular step are still limited in number [8]. In recent years, research reports on black tea rolling have mainly focused on the effects of the rolling process on the physical properties and flavor quality of tea leaves, as well as the optimization and exploration of different rolling methods. Studies of the physical properties of tea leaves mainly consider mechanical properties [9,10], electrical properties [11], and microstructures [12]. The study of different rolling methods includes process improvement and parameter optimization of factors such as leaf feeding volume [13], rolling temperature [14], rolling time [7,8], rolling frequency, and rolling pressure [15,16].

Non-targeted metabolomics is an emerging field and can be defined as the comprehensive analysis of all metabolites in a particular sample without specifically targeting or measuring any particular metabolites [17]. This approach aims to provide a broad and unbiased view of the metabolic profile of a sample, allowing for the identification of unexpected biomarkers, metabolic pathways, and potential metabolite interactions [18]. In this study, we used non-targeted metabolomics technology combined with sensory evaluation to identify the flavor substances in black tea. We then integrated this information into a multivariate statistical analysis to systematically reveal the effect of rolling time on the flavor quality of black tea. This study aims to improve the understanding of the black tea rolling process and provide a theoretical basis and technological support to optimize the process of producing black tea.

## 2. Materials and Methods

### 2.1. Chemicals and Reagents

Ninhydrin, anthrone, concentrated sulfuric acid, potassium dihydrogen phosphate, disodium hydrogen phosphate, stannous chloride, folinol, ethanol, n-butanol, ethyl acetate, sodium bicarbonate, and cyclohexanone were analytically pure and purchased from China Pharmaceutical Group, Beijing, China and The Shanghai Chemical Reagent Company, Shanghai, China. Methanol was chromatographically pure and purchased from Thermo Fisher, and analytically pure ethyl hydroxy theophylline was purchased from Shanghai Yuanye Bio-Tech Co., Shanghai, China. Detailed information on chemicals and reagents is listed in Appendix A.

### 2.2. Tea Samples Preparation

The raw material used for this study was autumn fresh leaves (one bud with two or three leaves) of Echa No. 10, which were picked from the local tea plantation in Shengshui Village, Zhushan County, Shiyan City, Hubei Province, China. The experiment followed the general process of processing fresh leaves—withering, rolling, fermentation, drying, and making tea—to produce the black tea. The fresh leaves were first evenly stacked 3–4 times after arrival at the processing unit to make them evenly mixed, and then they were uniformly spread on the withering trough (6CWD-200, Zhejiang Green Peak Machinery Co., Ltd., Quzhou City, Zhejiang province, China) for 10 h with alternate hot- and cold-air withering (first cold-air withering for 3 h, then hot-air withering for 7 h). The cold-air withering temperature was 23 °C, the humidity 78–82%, the leaves’ spreading thickness was about 15 cm, and the temperature of the hot-air withering process was set at 32 °C. The leaves were tossed several times during the hot-air withering process to ensure uniform water loss until the water content of the fresh leaves was around 60%. Then, the withered tea leaves were put into the rolling machine (6CR-65, Zhejiang Green Peak Machinery Co., Ltd., Quzhou City, Zhejiang province, China) for maceration. Seven different rolling time parameters, namely RT0, RT20, RT40, RT60, RT80, RT100, and RT120 (See Appendix A), were set for rolling process, during which the rolling machine was set at a temperature of about 25 °C with constant pressure. After the end of each treatment, the rolled tea leaves were uncurled and put into the fermenting machine (6CFJ-100, Zhejiang Green Peak Machinery Co., Ltd., Quzhou City, Zhejiang province, China) at a temperature of 32 °C, relative humidity of 95%, and fermentation time of 200 min. The fermented leaves were first dried at 130 °C in a chain-dryer (6CTH-60, Zhejiang Green Peak Machinery Co., Ltd., Quzhou City, Zhejiang province, China) to reduce the moisture content to about 20%, before being cooled for 1 h and again dried at 110 °C to a final moisture content of about 5–7%. Thus, tea samples were prepared, of which 1/4 was stored in an air-tight container at room temperature for sensory evaluation, and the remaining part was stored at −20 °C for further chemical analysis.

### 2.3. Sensory Evaluation of Black Tea

Sensory evaluation was performed according to the National Standard of China (GB/T23776-2018). The sensory attributes were independently scored by 5 professional tea tasters on a 100-point scale, with the appearance of dry tea accounting for 25%, liquor color for 10%, aroma for 25%, taste for 30%, and infused leaves for 10%. All the members of the tea tasting group are professionals (3 men and 2 women aged 30–50), with profound evaluation experience, from the Department of Tea Science, Huazhong Agricultural University (IRB certificate No. HZAUHU-2020-0003). Initially, the tea samples were placed on different white plates to evaluate their appearance. Subsequently, 3 g of dry tea were steeped with 150 mL of boiled water in a white porcelain cup for 5 min. Next, the tea liquor was poured into a white porcelain bowl to evaluate the liquor color. The five evaluators were asked to smell each sample and assess its aroma. Afterwards, the same five evaluators were asked to taste each sample and assess it. Finally, the soaked tea leaves without water were placed on a black square plate to evaluate the infused leaf. Thus, the sensory attributes of the black tea samples were evaluated. The sensory evaluation data were analyzed using a comparative statistical method.

### 2.4. Determination of Physicochemical Indicators of Tea

Total tea polyphenol content was determined using the Folinol colorimetric method following the Chinese national standard (GB/T 8313-2008) [19]. In brief, a 0.20 g tea sample was weighed and placed in a 10 mL centrifuge tube. Then, 5 mL of 70% methanol at 70 °C was added, and the mixture was heated in a water bath for 10 min. After cooling, the solution was centrifuged at 3500 r/min for 10 min, and the supernatant was collected. This process was repeated once, and the combined supernatants were adjusted to 10 mL with distilled water. After membrane filtration, 1 mL of tea extract was diluted to 100 mL. Next, 1 mL of the diluted extract was mixed with 5 mL of 10% Folin–Ciocalteu reagent. After incubation and the addition of 4 mL of 7.5% sodium carbonate, the solution was kept at room temperature for 1 h. Finally, the absorbance was measured at 765 nm to determine the tea polyphenol content using gallic acid solution as a standard.

Free amino acid content was determined using the ninhydrin colorimetric method based on the Chinese national standard (GB/T 8314-2013) [20]. A tea sample weighing 1.50 g was added to 50 mL of boiling distilled water. The solution was then immersed in a boiling water bath for 30 min and shaken every 10 min. It was filtered with cotton into a 100 mL volumetric flask, and the residue was washed 2–3 times with a small amount of hot distilled water. After cooling, the volume was adjusted with distilled water. Tea extract (1 mL) was added to a 25 mL volumetric flask. Then, 0.5 mL of potassium phosphate buffer solution (pH = 8.0) and 0.5 mL of 2% Ninhydrin solution were added. The solution was boiled in a water bath for 15 min, before being cooled to room temperature. Water was added to 25 mL to dilute the solution. Absorbance was measured at 570 nm.

The total soluble sugar content was determined using the anthrone–sulfuric acid colorimetric method [21]. In brief, a 1.50 g tea sample was weighed, and 50 mL of boiled distilled water was added. The mixture was then immersed in a boiling water bath for 30 min, with shaking occurring once every 10 min. The solution was filtered using cotton in a 100 mL measuring flask. The residue was washed 2–3 times with a small amount of hot distilled water and cooled to room temperature, before being filled to the volume mark with distilled water. Then, 1 mL of tea extract and 4 mL of distilled water were measured. The mixture was vortexed well. Next, 0.5 mL of the mixture was added to 4.0 mL of anthrone–sulfuric acid (this action was carried out in an ice bath). The mixture was then placed in a boiling water bath for 7 min. Finally, the absorbance was measured at 620 nm.

The contents of theaflavins, thearubigin, and theabrownins were determined by using a systematic analysis method [22]. In a concise procedure, a 3.00 g tea sample was boiled in 125 mL of distilled water for 10 min, filtered, and cooled. A 25 mL test solution was mixed with 25 mL of ethyl acetate, resulting in separated layers. The ethyl acetate layer (2 mL) formed Solution A, and the aqueous layer, after treatment with 2 mL of saturated oxalic acid, became Solution D. Additionally, ethyl acetate (15 mL) was treated with 15 mL of 2.5% sodium bicarbonate to create Solution C. Another test solution (15 mL) was mixed with n-butanol (15 mL) to form Solution B. All solutions were adjusted to a final volume of 25 mL with 95% ethanol. These solutions (A, B, C, and D) were prepared for analysis using a quartz cuvette and 95% ethanol as a reference. The absorbance was measured at 380 nm.

### 2.5. Determination of Volatile Components of Tea

Extraction method: Solid phase micro-extraction (SPME) was performed by aging the DVB-PDMS extraction fibers in the GC inlet at 250 °C for 1 h. A crushed tea sample weighing 1.00 g was added to a 20 mL headspace vial along with 5 mL of saturated Nacl aqueous solution at 100 °C and 0.5 mL (0.1 μL/mL) of cyclohexanone as the internal standard (IS). The extraction head was then inserted into the headspace vial and placed in a water bath at 60 °C for 60 min, before being inserted into the GC inlet and desorbed at 250 °C for 5 min.

Analytical method: Gas chromatography-mass spectrometry (GC-MS) was used, and the separation was performed using a DB-5MS column (30 mm × 0.25 mm × 0.22 μm, Agilent, Santa Clara, CA, USA). High-purity helium was used as the carrier gas at a flow rate of 1.0 mL/min. The column was initially operated at 45 °C, with a ramp rate of 7.0 °C/min to 80 °C. The ramp rate was changed to 2.0 °C/min and continued to 90 °C, where it was maintained for 2 min. It was then further increased to 100 °C at 3.0 °C/min and maintained for 2 min. The temperature was then raised to 130 °C at 3.0 °C/min and retained for 2 min, and, finally, it was increased to 230 °C at 10 °C/min for 5 min. The mass spectra obtained from GC-MS analysis were used to characterize the volatile components using NIST 14 and the retention index (RI). Relative quantification was performed using the internal standard method as follows: volatile component content to be measured = (volatile component peak area to be measured × internal standard content)/internal standard peak area.

### 2.6. Determination of Non-Volatile Components of Tea

Extraction method: The extraction of non-volatile metabolites was carried out in reference to the method of Xinlei et al. [23], albeit with some adjustments. Dried tea samples were taken and ground well in liquid nitrogen. Next, 150 mg of dried tea powder was weighed into a 20 mL volumetric flask and extracted with 7.5 mL of 75% (*v*/*v*) methanol solution containing 150 μL of etheophylline as an internal standard (etheophylline: CAS 519-37-9, weigh 1 mg, add 4 mL of 75% methanol (*v*/*v*) at 250 μg/mL) in a hot water bath (70 °C) for 30 min and then cooled to room temperature. The extract was transferred to a 10 mL centrifuge tube and centrifuged at 5000 rpm for 3 min. The supernatant was filtered through a 0.22 μm membrane, placed into a brown feed bottle, sealed with a sealing film and stored at −20 °C for further analysis.

Analytical methods: Metabolomic analysis was performed using ultra-performance liquid chromatography (UHPLC, Infinity 1290, Agilent, Santa Clara, CA, USA) in tandem with quadrupole time-of-flight mass spectrometry (Q-TOF/MS, Q-TOF 6520, Agilent, Santa Clara, CA, USA) using a Zorbax Eclipse Plus C18 column (100 × 2.1 mm, 1.8 μm, Agilent, Santa Clara, CA, USA). The chromatographic conditions used were as follows: mobile phase A was 0.1% (*v*/*v*) formic acid in water, and mobile phase B was methanol, following a gradient elution procedure.

The gradient elution procedure was as follows: 0–4 min, 10–15% B; 4–7 min, 15–25% B; 7–9 min, 25–32% B; 9–16 min, 32–40% B; 16–22 min, 40–55% B; 22–28 min, 55–95% B; 28–30 min, 95% B; 30–31 min, 95–10% B; and 31–35 min, 10% B. The mass spectrometry was performed in ESI+ mode with a capillary voltage of 3.5 kV; a drying gas temperature and a flow rate of 300 °C and 8.0 L/min, respectively; a spray pressure of 3.5 psi; a sheath gas temperature and a flow rate of 350 °C and 11.0 L/min, respectively; and a scan range of 100–1200 Da. These conditions facilitate the qualitative analysis of compounds, and the three collision energies set for the controlled sample injection automatic secondary (Auto MS/MS) scans were 10 V, 20 V, and 30 V, respectively.

### 2.7. Statistical Analysis

Data are presented as mean ± standard deviation. GC-MS and UHPLC-Q-TOF/MS raw data were analyzed using Thermo Xcalibur 3.1.66.10 (XReport, and XDK reference) and NIST MS Search 2.2 software. Multivariant statistical analysis and one-way ANOVA with an LSD post hoc test were performed using SPSS 20.0 software, and statistical significance was considered to be achieved when *p* < 0.05. Principal component analysis (PCA) and orthogonal partial least square discriminant analysis (OPLS-DA) were performed using SIMCA 14.1 software. TB Tools was used to produce the heatmaps, and histograms were produced using GraphPad Prism 9.2.0.

## 3. Results and Discussion

### 3.1. Effects of Rolling Time on the Sensory Quality of Black Tea

The sensory evaluation of black teas with different rolling times revealed significant differences in the sensory quality of black teas produced at various rolling times (see Table 1). As the rolling time increased, the firmness and evenness of the black tea’s shapes gradually improved. The color of the dry tea became darker, and the color of the tea broth changed from red-yellow to a rich red. The aroma gradually lost its greenish scent and revealed floral and sweet aromas. The taste became more mellow and fresh, and the color of the leaf base became more even and bright. The overall sensory score showed a trend of initial increase and then decrease, with the sensory quality of black tea improving between 0 min and 100 min of rolling. In particular, when the 60 min treatment was regarded as the cut-off point, the greenish and astringent taste of black tea faded and gradually developed into a fresh and refreshing taste. The results of the sensory evaluation showed that a moderately long rolling time was beneficial to the formation of black tea shape and inner quality, with the best overall sensory scores being obtained for 80 min and 100 min of rolling.

### 3.2. Effect of Rolling Time on Physicochemical Composition of Black Tea

Comparing the physicochemical compositions of black tea under different rolling time treatments (see Table 2), it was found that the contents of tea polyphenols and free amino acids showed a significant decreasing trend during the rolling process. The contents of soluble sugars peaked at 20 min of rolling, and there was no significant difference between 40 min and 80 min of rolling. However, the contents significantly increased at the later stage of rolling (100 min to 120 min). The theaflavin and thearubigin contents showed a trend of initial increase and then decrease during the rolling process, and both of them reached their peak at 80 min of rolling. Meanwhile, the theabrownin content showed a significant increase as the degree of rolling increased. The experimental results indicate that the twisting process promotes the transformation and formation of black tea flavor substances.

Moreover, the initial stage of rolling (0–20 min) showed drastic changes in the contents of all the compounds, thereby showing this period to be the most critical of all. Hence, utmost care and precautions should be taken into consideration during this period of the rolling process to obtain high-quality black tea. Furthermore, since tea polyphenols are the main source of bitterness and astringency in tea broth [24], while theaflavins not only have good biological activity but also a bitter and astringent taste [25], a moderately long rolling time can reduce the bitter and astringent taste of black tea.

### 3.3. Effects of Rolling Time on Volatile Components of Black Tea

Aroma is the main influencing factor involved in determining the quality of finished tea. In this study, the volatile components of tea with different rolling times were analyzed and identified by using HS-SPME-GC-MS technology. A total of 83 common aroma components (see Appendix A) were screened out after NIST14, including 24 alcohols, 23 esters, 14 aldehydes, 10 hydrocarbons, 5 ketones, 4 acids, and 3 phenols. By using the 83 shared volatile components as the dependent variable and 7 different rolling time treatments as the independent variables, the OPLS-DA analysis showed the effective differentiation of the black tea samples made with different rolling times. As shown in Figure 1A, the sample points of the aroma components of the 0-min rolling samples fell in the fourth quadrant alone, which was far away from the sample points of the other treatments, indicating that the endogenous volatile components of the withered leaves significantly changed after the rolling treatment. The results showed that the fit index of the independent variable (R2X) was 0.909, the fit index of the dependent variable (R2Y) was 0.924, and the predictive index of the model (Q2) was 0.408. After 200 substitution tests (Figure 1B), the point of intersection of the regression line and the vertical axis of Q2 were less than zero, which validated the model.

The results of GC-MS analysis showed that the total amount of aroma substances in black dried tea significantly increased after the beginning of rolling and remained stable in the subsequent rolling stages. This result indicates that the initial stage of rolling (0–20 min) is an important point for the generation of a large number of volatile substances.

The rolling process is an uncontrolled fermentation process. During this process, a large number of tea cells are broken, causing aroma precursor substances to rush into the cells. These substances then undergo enzymatic oxidation, resulting in the formation of numerous aroma substances [26]. The total amounts of aroma substances increase as the rolling time is prolonged. Of the different rolling times, RT80 has the highest total amount of aroma substances, while RT0 has the lowest. This difference is mainly observed in the presence of alcohols, aldehydes, carboxylic acids, and ketones (shown in Figure 2).

The contents of alcohols were the highest of all total aroma substances (50.93–56.42%). The contents of alcohols significantly increased between 0 min and 20 min of rolling, but as the rolling time continued to increase, the total amounts of alcohols did not significantly change. The alcohols present in higher contents (≥5% of the total amount of aroma) included geraniol, phenylethanol, linalool, linalool oxide, and benzyl alcohol, which accounted for about 80% of the total amount of alcohols. The content of geraniol (soft rose aroma) accounted for 9.53~13.35% of the total aroma, and it showed an increasing trend in tandem with the rolling process, with an increase of 43.82~64.91%. The content of linalool (lily-of-the-valley aroma) was second only to geraniol, and it showed a pattern of first increasing and then decreasing in tandem with the rolling process. It reached a peak at 40 min of rolling and gradually decreased after 80 min of rolling. The content of benzyl alcohol (rose aroma) also increased during the twisting process. Linalool and its oxides are the main substances involved in determining the floral aroma of black tea [27]. Geraniol and linalool are isomers; thus, it is judged that these floral alcohols make an important contribution to the formation of the aroma quality of black tea during processing. The initial stage of rolling (0~20 min) is an important node for its formation in large quantities.

The total amount of ester compounds measured was second only to alcohols, and their contents showed an overall decreasing trend. The substances for which ester compounds accounted for ≥5% of the total aroma were methyl salicylate, foliate caproate, trans-2-hexenyl caproate, dimethyl phthalate, hexyl caproate, and foliate isopentanoate. Among them, methyl salicylate (mint, wintergreen odor) had the highest content among the esters, and its significantly content increased between 0 min and 20 min of rolling. However, there was no obvious regular change in its content after 20 min of rolling. In addition, the contents of foliate caproate (fruity aroma), hexyl caproate (peach odor), and foliate isovalerate gradually decreased during the rolling process. The content of trans-2-hexenyl caproate also decreased after 80 min of rolling, whereas the content of iso-octyl salicylate significantly increased at 60 min of rolling, before returning to a relatively stable level after 60 min. It can be observed that the ester compounds with fruit aroma characteristics did not exhibit a clear pattern of change in tandem with the increase in the degree of rolling.

The aroma characteristics of aldehydes include fresh, grassy, floral, and fruity aromas, which significantly contribute to the aroma of black tea [28]. The total number of aldehydes increased as the rolling time increased. The aldehydes with relative contents ≥10% were phenylacetaldehyde, cis-citraldehyde, and β-cyclocitraldehyde. These were followed by citral, E-2-octenal, and (E,E)-2,4-heptadienal, which had a shares ≥5% in the total aroma. Phenylacetaldehyde (Suzuran) was present in the highest content, which increased by 111.65% between 0 min and 20 min of rolling. However, there was no obvious change pattern after 20 min of rolling. The contents of citral (fruity flavor) and benzaldehyde (bitter almond flavor) also greatly increased after the beginning of rolling (0 min), but the changes in their contents were relatively small in the middle and latter stages of rolling. On the other hand, the content of β-cyclocitral (lemon odor) showed a decreasing trend during the rolling process. The increase in the total amount of aldehydes implies that there are favorable factors promoting the synthesis of aldehydes in black tea during the rolling stage.

Additionally, five ketones were identified, with β-violetone being the most abundant, accounting for approximately 1.5% of the total volatile matter content. Its content showed an increasing trend in correlation with the prolongation of rolling time. Furthermore, this study detected 17 other compounds, consisting of 10 hydrocarbons, 4 acids, and 3 phenols. The content of limonene was the highest among these aroma substances, accounting for about 5% of the total aroma substances. However, its content did not significantly change with the extension of the rolling time.

Based on the results of the GC-MS analysis, it was observed that volatile compounds were generated in large quantities during the initial 20 min of the black tea rolling process. These compounds continued to increase; however, some of the characteristic aroma components of black tea decreased after 80 min of rolling. Sensory evaluation also showed that the sweet and floral aroma of the tea significantly increased at first, before gradually decreasing up to 80 min during the rolling process. The aromatic compounds, such as alcohols (Linalool and its oxides, geraniol, etc.) and aldehydes (phenylacetaldehyde, cis-citraldehyde, and β-cyclocitraldehyde), which are responsible for the floral and fruity aroma, exhibited similar trends during the process of rolling black tea. Therefore, the results of the GC-MS analysis were essentially consistent with the findings of the sensory evaluation.

### 3.4. Effects of Different Rolling Times on the Non-Volatile Components of Black Tea

Tea polyphenols, pigments, amino acids, soluble sugars, and flavonoids form the basis of black tea flavor characteristics. A total of 60 non-volatile compounds were identified from the samples using UPLC-Q-TOF/MS (see Appendix A). These included 3 alkaloids, 4 theaflavins, 8 organic acids, 15 flavonoids and flavonoid glycosides, 7 aroma glycosides, 16 amino acids, and 7 catechins. The OPLS-DA analysis (Figure 3A) yielded a fit index R2X of 0.992 for the independent variables, a fit index R2Y of 0.736 for the dependent variables, and a model prediction index Q2 of 0.339. The reasonableness of the OPLS-DA analysis was demonstrated through 200 permutation tests using the cross-validated model (Figure 3B). The sample sites of non-volatiles from RT0 were exclusively located in the fourth quadrant, while the other rolling time treatments, applied further away from its position, were arranged in rows base on the time gradient. This finding suggests that the non-volatile constituents underwent noticeable regular changes during the black tea rolling process.

In the samples treated with different rolling times, the relative total amount of non-volatile substances was highest in RT0 and lowest in RT120. As shown in Figure 4, the alkaloid content in the finished tea significantly increased with longer rolling times. The theaflavin content initially increased and then decreased. The amino acid content remained stable and did not show a clear pattern of change. In contrast, catechin, organic acid, flavonoid, flavonoid glycoside, and aromatic glycoside contents were all significantly reduced.

The content of alkaloids increased in tandem with the increase in the depth of rolling, especially in the early stage of rolling (0–20 min), with an increase of 13.6% recorded. Caffeine, the most abundant alkaloid, accounted for about 37.75–53.52% of all non-volatile components, and its content also increased in line with the rolling process. Caffeine is an important flavoring substance in tea, and the complex formed by hydrogen bonding with theaflavin has a fresh and refreshing taste [29].

Theaflavin is the main antioxidant component in black tea, and it can effectively scavenge free radicals, chelate metal ions, and inhibit pro-oxidant enzyme activities [30]. It also determines the mellowness and freshness of the black tea flavor [31], playing a key role in the formation of good black tea quality. During the rolling stage, the total amount of theaflavins initially increased and then decreased, especially between 0 min and 20 min of rolling, with an increase of 28.89–42.22% recorded. After 80 min of rolling, the amount of theaflavins decreased, which may be attributed to the further conversion of theaflavins into thearubigins and other substances. Some studies have shown that theaflavins and their gallic acid esters can combine with caffeine to generate theaflavins. When the temperature of the tea broth decreases, it causes these components to settle, thus playing a role in alleviating the bitter flavor of caffeine and affecting the overall taste of the tea broth [32].

Flavonoids are a class of low-molecular-weight polyphenolic compounds [33]. Flavonoids in black tea mainly consist of isoflavones, flavanones, and flavonols. Flavonols are often combined with sugars to form glycosides and exist as 3-O-glycosides [34]. The flavonoids present in tea include poncirin (glycoside), yangmeiin (glycoside), quercetin (glycoside), and sanguinarine (glycoside). Flavonoid glycosides contribute to both the color and taste of tea broth, providing an astringent sensation that causes dryness and crinkling in the mouth [35]. A total of 4 flavonols and 11 flavonoid glycosides were isolated and identified in the experiment. The number of identified flavonoid glycosides was significantly higher than that of flavonols, accounting for about 70% of the total flavonoids and flavonoid glycosides. Furthermore, their overall contents gradually decreased as the rolling time was prolonged. Notably, the flavonoid glycosides with higher contents included populin-3-O-galactoside, quercetin 3-O-rutinoside, quercetin, and isomonycin-2″-O-arabinoside. Among them, the contents of populin-3-O-galactoside and quercetin 3-O-rutinoside significantly decreased with the deepening of the rolling degree, whereas the contents of quercetin and isoosmetin-2″-O-arabinopyranoside increased during the rolling process. The results indicated that the black tea rolling process could reduce the astringent taste produced by flavonoid glycosides and enable the black tea flavor to be more harmonious and palatable [36].

During tea processing, organic acids are mainly produced through the decomposition of sugars. These organic acids are closely related to the flavor of tea broth. For example, gallic acid enhances the freshness of monosodium glutamate, while oxalic acid affects the sourness of black tea [37]. The total measured organic acids showed a decreasing trend during the rolling process. Among the organic acids, mangiferic acid had the highest content, but its content decreased by 15% between 0 min and 20 min of rolling. Mangiferic acid is a key intermediate product in the mangiferic acid pathway, which is the main pathway for the microbial synthesis of aromatic compounds and their derivatives. Mangiferic acid can generate phenylalanine under enzyme catalysis. In this experiment, a significant increase in phenylalanine content was detected in the rolling stage (an increase of 217.65%). The phenylalanine metabolic pathway is one of the most important pathways in the secondary metabolism of the tea tree, which can synthesize aromatic substances such as phenylethanol and benzyl alcohol [38]. Therefore, it is determined that rolling can accelerate the metabolism of mangiferic acid and provide precursors for the synthesis of some tea aroma components. In addition, the contents of salicylic acid, tartaric acid, and chlorogenic acid significantly decreased during the rolling process. Gallic acid significantly increased after the beginning of rolling, before decreasing after 60 min of rolling, whereas the contents of α-ketoglutaric acid, succinic acid, and pyruvic acid did not show any obvious pattern of change.

Volatile aroma substances are present in fresh leaves in the form of glycosides, which are released via hydrolysis enabled by endogenous enzymes during black tea processing [39]. In this study, a total of four glucosides and three primrose glycosides were measured, and all of these glycosides were hydrolyzed during the rolling process, resulting in a significant decrease in content. Among them, phenyl ethanol glucoside had the highest content, and its content decreased dramatically between 0 min and 20 min of rolling, with a reduction range of 91.41–96.93% recorded. In addition, the content of primrose glycosides significantly reduced in tandem with the prolongation of rolling time. The large number of broken tea cells at the rolling stage provided a reaction opportunity for primrose glycosidase on the cell wall and primrose glycosides in the vesicles. This led to the release of ligands in the glycosides during the hydrolysis reaction, resulting in a sharp decrease in the content of primrose glycosides throughout the rolling process [40]. During the rolling stage, the content of sakuragulosides, glucosides of geraniol, benzyl alcohol, and phenylethanol significantly decreased due to hydrolysis. Even in the middle and latter stages of rolling, the contents of sakuragulosides of cyclohexanol and geraniol sakuragulosides could no longer be detected. The finding partially explains the increase in the contents of the volatile constituents of corresponding alcohols in the finished teas.

The free amino acids in tea leaves are the key substances that determine the freshness and sweetness of black tea. According to the taste characteristics of different amino acid monomers, they can be classified into four categories: fresh amino acids, sweet amino acids, bitter/umami amino acids, and astringent amino acids [41]. In this study, the highest proportion of fresh amino acids was detected, and its content increased in line with the prolongation of the rolling time. Theanine had the highest content of all the fresh amino acids, accounting for about half of the total amino acids. It has a caramel aroma and fresh taste, making it a characteristic amino acid component of tea. This amino acid has become an important factor involved in evaluating the quality of tea [42]. The total amount of bitter amino acids showed a significant decreasing trend as rolling continued, except for phenylalanine content, which significantly increased during the rolling process. Some studies have shown that the phenylalanine content has a significant positive impact [43]. Phenylalanine metabolism is an important secondary metabolic pathway of the tea tree, and all substances with a phenylpropanoid skeleton are generated through this pathway. The phenylalanine metabolic pathway can synthesize various pigment components in plants, and many key signaling substances (such as salicylic acid, abscisic acid, etc.) involved in plant development are mainly produced through this pathway [44]. In addition, tryptophan has a positive effect on the formation of black tea aroma quality, which is a precursor substance for the synthesis of floral volatile components such as indole. The decrease in tryptophan content during rolling may be related to its involvement in the synthesis of secondary products such as indole [45]. Moreover, the total amount of sweet amino acids gradually decreased with the increase in the rolling time, while the contents of astringent amino acids remained more stable during the rolling process. In connection with the results of sensory evaluation, it can be speculated that free amino acids are important substances that contribute to enhancing the fresh taste and reducing the bitterness of black tea.

Catechins are the main components of tea polyphenols, and the catechin monomers mainly consist of eight types, namely EGCG, EC, GC, C, EGC, GCG, ECG, and CG; among these types, EGCG, ECG, GCG, and CG are complex catechins, also known as ester-type catechins, while C, EC, EGC, and GC are simple catechins, also known as non-ester-type catechins. Ester catechins have strong bitterness and astringency, and they are the main substances that lead to the bitter and astringent flavor of tea, while non-ester catechins have weaker astringency and a pleasant aftertaste [46]. The contents of all seven catechin monomers detected significantly decreased with the prolongation of the rolling time. The total amount of ester catechins (EGCG, ECG, GCG) was higher than that of non-ester catechins (EC, EGC, GC, C), which accounted for 53.65–71.46% of the total amount of catechins, and the total amount of ester catechins decreased by 87.69% during the rolling process. The highest amount of all the ester catechins was found to be ECG, whose content significantly decreased during the rolling process, with a decrease of 79.84% recorded. ECG has a low astringency threshold [47] and high flavor intensity [48]. The overall decrease in non-ester catechins was greater than that of ester catechins, with a decrease of 93.47% recorded. During the rolling stage, a large amount of water was lost from the tea cells, which gradually increased the activities of poly PPO and POD. Also, the destruction of the cellular structure broke the isolation between catechins and oxidative enzymes, making it easier for oxygen to enter the cells. As a result, the catechins accelerated oxidation and progressed to produce theaflavin, leading to a significant reduction in the catechin content [46,49]. Therefore, a moderate extension of the rolling time can significantly reduce the catechin content, thereby attenuating the bitter and astringent taste of black tea and improving the overall flavor quality of the tea broth.

The results of the sensory evaluation showed that the freshness, strength, and sweetness of the tea gradually increased during the rolling process. The tea made during 80–100 min of rolling was considered fresh and strong in taste. The analysis of differential metabolites revealed that amino acids and polyphenols played crucial roles in determining the taste quality of the tea [50]. The total contents of free amino acids and tea polyphenols in black tea showed a downward trend during the rolling process, which was consistent with the results obtained via LC-MS.

It is important to give valuable attention to the rolling time parameter during the industrial manufacturing of Gongfu black tea for quality improvement. This study provides a theoretical basis for optimizing the process of Gongfu black tea made from a bud and two leaves or higher mature leaves. It also helps to improve the quality of other black teas made from delicate leaves (single buds). However, the rolling time does not solely control the quality of black tea. Therefore, it is recommended to conduct extensive research using metabolomic and transcriptomic approaches to understand the impact of the rolling process on the quality of black tea.

## 4. Conclusions

In this study, sensory evaluation, quantitative analysis, and a non-targeted metabolomics approach were employed to investigate the dynamic change in volatile and non-volatile metabolites due to the influence of rolling time during the processing of black tea. This study helps us to understand the quality formation of black tea during different stages of the rolling process. Sensory evaluation, GC-MS analysis for volatile components, and UHPLC-Q-TOF/MS analysis for non-volatile components showed that the overall quality of the black tea products obtained through rolling for 80–100 min was higher. On one hand, the shape was tighter and finer, while on the other hand, the content of phenylalanine significantly increased and was able to develop a better aroma, and the total amount of catechins was further reduced, leading to a more mellow flavor. The degradation of GBVs also promoted the generation of endogenous aroma components in black tea. Moreover, during Gongfu black tea production, delicate tea raw materials such as single buds are often subjected to light rolling for about 20 min to maintain the intact and beautiful shapes of the tea leaves. Our experimental results also confirm that a short period (20 min) of light rolling can cause significant changes in the tea’s taste substances. This includes a notable increase in alkaloids and flavonoids, while the contents of various catechins, amino acids, and glycosides significantly decrease. These findings fully demonstrate the importance of the rolling process.

In conclusion, our study examined the changes in volatile and non-volatile substances during the rolling process from the perspective of metabolomics. The results proved that the metabolic substances of tea leaves continuously changed during the rolling process, and there were significant differences between the tea leaves before and after 20 min of rolling. For commonly used raw materials with one bud and two leaves and higher maturity, an appropriate rolling time of 80–100 min was determined. It was found that a rolling period that was too long did not contribute to the aroma and taste quality. Further extensive research into other rolling parameters is required in order to understand the impact of the rolling process on the molecular mechanism of black tea quality formation. The results of this study could have a guiding effect on the selection of rolling parameters in the processing of Gongfu black tea, which requires a complete appearance and high aroma and flavor.

## Figures and Tables

**Figure 1 foods-13-00325-f001:**
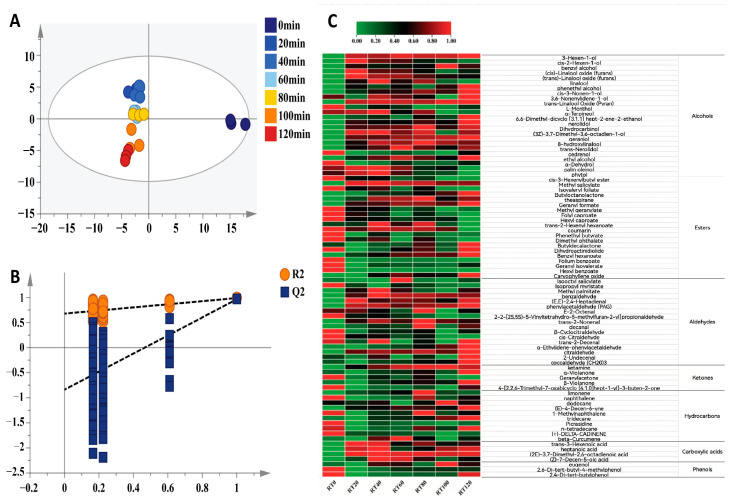
Multivariate statistical analysis of volatile metabolites of black tea treated with different rolling times. (**A**) OPLS-DA score plot, (**B**) cross-validation, and (**C**) heatmap of the contents of volatile components in black tea treated with different rolling times.

**Figure 2 foods-13-00325-f002:**
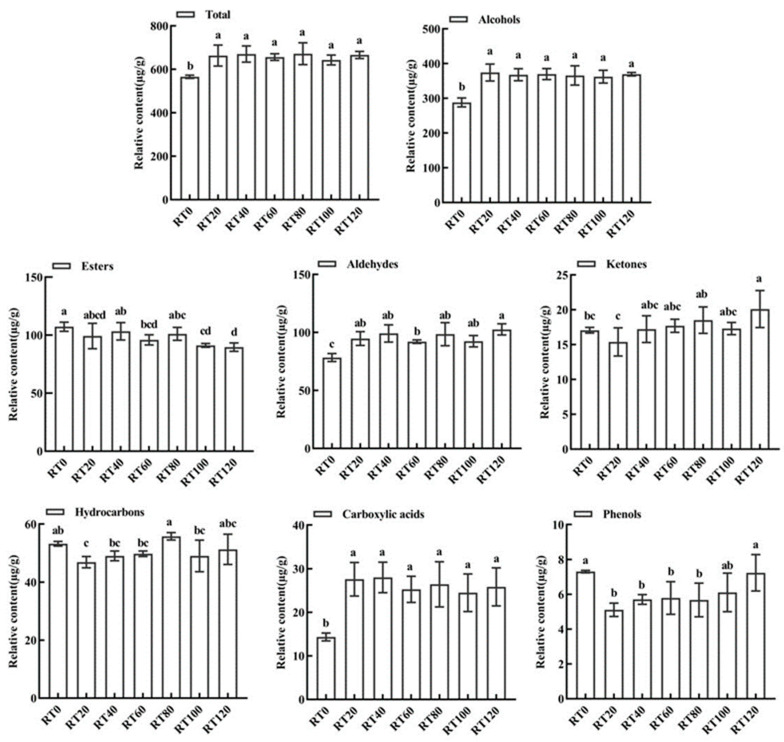
Effects of different rolling times on the main volatile compound types of black tea (µg/g). Different letters in the same sub-graph indicate significant differences (*p* < 0.05).

**Figure 3 foods-13-00325-f003:**
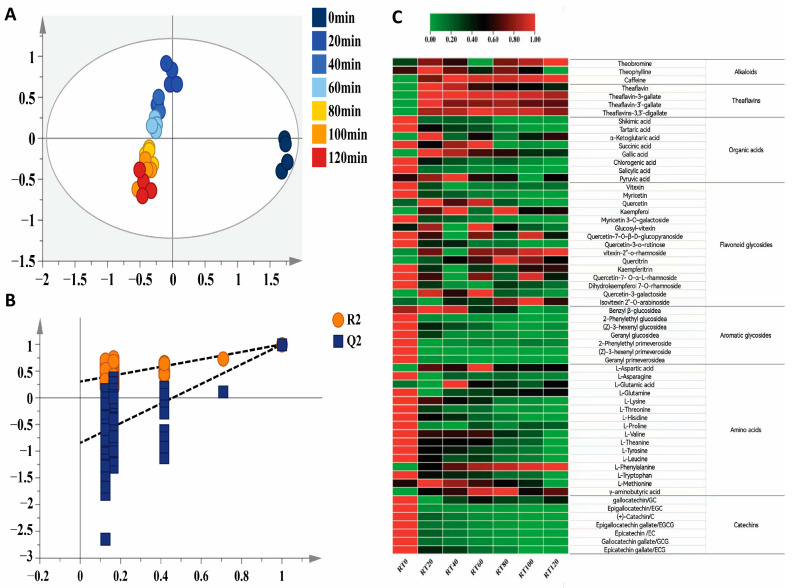
Multivariate statistical analysis of non-volatile metabolites of black tea treated with different rolling times. (**A**) OPLS-DA score plot, (**B**) cross-validation, and (**C**) heatmap of the contents of non-volatile components in black tea treated with different rolling times.

**Figure 4 foods-13-00325-f004:**
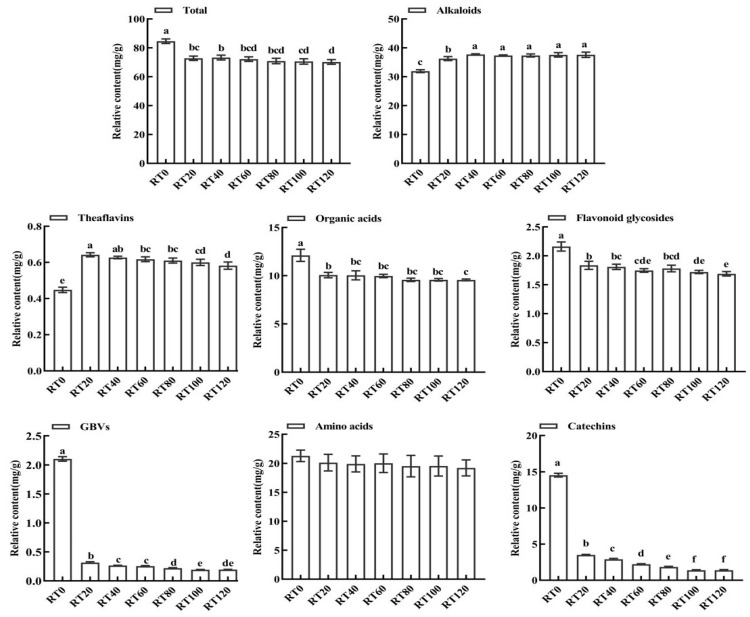
Effects of different rolling times on the main non-volatile compound types of black tea (mg/g). Different letters in the same sub-graph indicate significant differences (*p* < 0.05).

**Table 1 foods-13-00325-t001:** Sensory evaluation results of black tea treated with different rolling times.

Sample	Appearance (25%)	Liquor Color (10%)	Aroma (25%)	Taste (30%)	Infused Leaf (10%)	Total Score
RT0	The tea strips are in natural form, yellow-green and dark in color, with many brown flakes, and not well blended, and the buds are covered with white hairs	Red, yellow, bright	Mixed, green, dull	Bitter, slightly green	Still tender, yellowish-green, mixed flowers, not well blended	69.63 ± 0.3 f
	65.33 ± 0.57 f	76.67 ± 0.57 e	67.67 ± 0.57 e	73.5 ± 0.5 g	67.5 ± 0.5 e	
RT20	Coarser, still well balanced, brownish withered, more hairy	Orange-red, bright	Mixed, green, slightl dull	Slightly astringent and greenish	Still tender, yellowish-red, mixed flowers, not well-mixed	76.14 ± 0.21 e
	75.17 ± 0.28 e	84.67 ± 0.57 d	71.67 ± 0.57 d	78.53 ± 0.5 f	74.5 ± 0.5 d	
RT40	Still tightly knotted, still well-balanced, brownish color, slightly withered, more hairs	Still red, Bright	Slightly mixed, slightly green, slightly dull	Slightly greenish	Still tender, yellowish-red, mixed flowers, not well-mixed	79.5 ± 0.31 d
	77.33 ± 0.57 d	88.5 ± 0.5 c	76 ± 0 c	81.5 ± 0.5 e	78.67 ± 0.57 c	
RT60	Tightly knotted, still well-balanced, with many brown flakes, brownish color, with hairs	Red and bright	Sweet, floral	Mellow, still fresh	Still tender, still red, bright, still toned	85.82 ± 0.14 c
	81.17 ± 0.28 c	90.83 ± 0.28 b	88.17 ± 0.28 ab	86.17 ± 0.28 c	85.5 ± 0.5 b	
RT80	Tightly knotted, still well-balanced, brownish color, with hairs	Red, thicker, brighter	Sweet, floral	Strong and fresh	Still tender, still red, bright, still toned	87.32 ± 0.18 a
	82.67 ± 0.28 b	92.83 ± 0.28 a	89 ± 0.5 a	88.33 ± 0.28 a	86.17 ± 0.28 ab	
RT100	Tightly knotted, still rounded, relatively even, still dark and moist, slightly hairy	Red, thicker, brighter	Sweet, floral	Strong, fresh	Still tender, still red, bright, still toned	87.32 ± 0.05 a
	84.67 ± 0.28 a	92.83 ± 0.28 a	88.33 ± 0.57 a	87.17 ± 0.28 b	86.4 ± 0.36 a	
RT120	Tightly knotted, rounded, well balanced, still dark and moist, slightly hairy	Thick, bright red	Sweet, floral	Rich liquor	Still tender, still red, bright, still toned	86.41 ± 0.21 b
	84.83 ± 0.28 a	92.83 ± 0.28 a	87.33 ± 0.57 b	85 ± 0.5 d	85.83 ± 0.28 ab	

Note: RT0 represents the rolled 0 min sample; RT20 represents the rolled 20 min sample; RT40 represents the rolled 40 min sample; RT60 represents the rolled 60 min sample; RT80 represents the rolled 80 min sample; RT100 represents the rolled 100 min sample; and RT120 represents the rolled 120 min sample. Different letters in the same column indicate significant differences between groups (*p* < 0.05).

**Table 2 foods-13-00325-t002:** Physical and chemical analysis results of black tea treated with different rolling times (%).

Sample	Tea Polyphenol	Free Amino Acids	Soluble Sugar	Theaflavin	Thearubigin	Theabrownin
RT0	13.28 ± 0.15 a	4.13 ± 0.03 a	3.02 ± 0.04 b	0.29 ± 0.016 e	3.97 ± 0.11 d	5.7 ± 0.06 g
RT20	11.81 ± 0.17 b	3.81 ± 0.06 bc	3.12 ± 0.06 a	0.42 ± 0.025 b	4.14 ± 0.07 bc	7.36 ± 0.09 f
RT40	11.6 ± 0.24 bc	3.81 ± 0.05 bc	3.01 ± 0.03 b	0.41 ± 0.015 bc	4.2 ± 0.09 ab	7.77 ± 0.1 e
RT60	11.26 ± 0.25 cd	3.83 ± 0.05 b	3.01 ± 0.04 b	0.42 ± 0.007 b	4.19 ± 0.1 ab	7.94 ± 0.19 d
RT80	11.01 ± 0.17 d	3.77 ± 0.04 c	3.02 ± 0.07 b	0.46 ± 0.006 a	4.28 ± 0.08 a	8.42 ± 0.07 c
RT100	10.57 ± 0.25 e	3.55 ± 0.03 d	3.09 ± 0.02 a	0.38 ± 0.01 d	3.98 ± 0.06 d	8.56 ± 0.08 b
RT120	10.53 ± 0.62 e	3.56 ± 0.03 d	3.08 ± 0.06 a	0.4 ± 0.018 cd	4.04 ± 0.13 cd	8.78 ± 0.07 a

Note: different letters in the same column indicate significant differences (*p* < 0.05).

## Data Availability

Data are contained within the article and Appendix A.

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
