# Peer review of "Non-Targeted Metabolomics Reveals the Effects of Different Rolling Methods on Black Tea Quality"

_foods, 2024, doi:10.3390/foods13020325_

Round 1
Reviewer 1 Report
Comments and Suggestions for Authors
The manuscript presents an intriguing exploration of the impact of rolling time on the quality parameters of black tea, employing a combination of non-targeted metabolomics, sensory evaluation, and multivariate statistical analysis. However, there are certain aspects that need further attention:
1. While the manuscript discusses the findings from GC-MS and UHPLC-Q-TOF/MS analyses, it lacks specific details about the identified aroma components, catechins, flavonoids, and theaflavins. More specific information would enhance the scientific rigor and applicability of the study.
2. Elaborate on the statistical methods used for data analysis. Specify the significance thresholds and discuss the statistical reliability of the observed trends and differences.
3. Provide more information on the sensory evaluation methodology. Include details on the panel, criteria for evaluation, and statistical analysis of sensory data.
4. Connect the results of GC-MS and UHPLC-Q-TOF/MS analyses with sensory evaluations. How do the identified compounds correlate with sensory attributes?
5. Discuss the broader implications of the findings for the tea industry. Consider addressing potential limitations and suggesting directions for future research. Also, clarify if the results are specific to Gongfu black tea or if they can be generalized to other types.
6. The conclusion emphasizes the optimal rolling time, but it would be beneficial to explicitly state the practical recommendations based on the findings. What does this mean for tea processing practices?
Comments on the Quality of English LanguageModerate editing of English language required
Reviewer 2 Report
Comments and Suggestions for Authors
The manuscript is well designed with good presentation of result. Some comments should be included in the text as follows:
1- In method section, it should include the details of analysis for the part "Determination of physicochemical indicators of tea
2- In table 1, there is abbreviation RN that is not defined the complete form. Totally, some abbreviation in the text is not defined. It should revise this part.
3- I haven't found the supplementary file
4- It should more explain about the section of sensory evaluation, the type of this analysis "is that consumer or product oriented" and the kind of statistical analysis for this part
Good luck
Reviewer 3 Report
Comments and Suggestions for Authors
In the paper "Non-targeted metabolomics reveals the effects of different rolling methods on black tea quality," a great many distinctions are presented. A reasonable direction to control so many variables seems to be the use of chemometrics. The authors made careful use of Non-targeted metabolomics. Techniques based on the analysis of large data sets are increasingly common to describe the experiment performed and allow to mature some relationships unavailable with classical data analysis.
The authors describe and discuss well about sensory evaluation and indicators of black tea. The authors thoroughly describe about volatile components and non-volatile components of tea and their influence on the quality of black tea.
However, the reviewer has some observations, please take them into account when editing the manuscript:
Line 92
What were the parameters of hot and cold air withering (temperature, moisture)?
Line 106-107
What was the purpose of storing the samples at -20°C? The tea was dry.
Jak wyglÄ…daÅ‚ proces przygotowywania próbek od wyjÄ™cia z temperatury -20°Cdo momentu parzenia.
What was the process of preparing the samples from removal from the -20°C temperature until the tea was brewed.
Have you measured changes in the moisture content of tea samples, frozen and ready to brew ?
Line 209
Description in Table 2, Soluble describe Sugars or Soluble Sugars.
Round 2
Reviewer 1 Report
Comments and Suggestions for Authors
Accepted for publication
Comments on the Quality of English LanguageMinor editing of English language required